# Observation of discrete-light temporal refraction by moving potentials with broken Galilean invariance

Chengzhi Qin[1,5], Han Ye[1,5], Shulin Wang[1,5], Lange Zhao[1], Menglin Liu[1], Yinglan Li[1], Xinyuan Hu[1], Chenyu Liu[1], Bing Wang [1]✉, Stefano Longhi [2,3]✉ & Peixiang Lu [1,4]✉

Refraction is a basic beam bending effect at two media's interface. While traditional studies focus on stationary boundaries, moving boundaries or potentials could enable new laws of refractions. Meanwhile, media's discretization plays a pivotal role in refraction owing to Galilean invariance breaking principle in discrete-wave mechanics, making refraction highly moving-speed dependent. Here, by harnessing a synthetic temporal lattice in a fiber-loop circuit, we observe discrete time refraction by a moving gauge-potential barrier. We unveil the selection rules for the potential moving speed, which can only take an integer $v = 1$ or fractional $v = 1/q$ (odd $q$) value to guarantee a well-defined refraction. We observe reflectionless/reflective refractions for $v = 1$ and $v = 1/3$ speeds, transparent potentials with vanishing refraction/reflection, refraction of dynamic moving potential and refraction for relativistic Zitterbewegung effect. Our findings may feature applications in versatile time control and measurement for optical communications and signal processing.

Refraction, i.e., the bending of light beam at the interface between two continuous media with different refractive indices, is a basic and ubiquitous phenomenon in optics known for centuries and discussed at length in physics textbooks. Quite recently, the concept of refraction has been extended to temporal interfaces[1–4], and a surge of interest is devoted to studying wave dynamics in time-varying systems[5,6], such as time-varying metamaterials, surfaces[6] and photonic time crystals[6–8], enabling extreme control of electromagnetic waves in unprecedented ways. Early examples include wave propagation in moving continuous media and scattering by moving interfaces[9–12], where important phenomena including the Fizeau drag effect, i.e., light speed change by medium motion[13–15], and optical analogs of relativistic effects, like the event horizon or Hawking radiation[16,17], can be observed. Moving potentials provide another important class of time-varying media, which have been considered in atomic and optical systems to induce topological Thouless pumping[18,19] and for observing quantum time reflection and refraction of ultracold atoms[20].

So far, most studies on temporal refractions have considered continuous media. However, in many integrated photonic structures light behavior is discretized in space and the effective light speed on the lattice, known as the group velocity as determined by inter-site hopping rate, is non-relativistic, leading to an effective description of light dynamics in terms of the discrete Schrödinger equation[21]. Refraction can also occur at the interface between two discrete lattices, like optical waveguide arrays[22,23], photonic crystals[24,25], and metamaterials[26,27], obeying a different Snell's law[28]. Interestingly, if

[1]Wuhan National Laboratory for Optoelectronics and School of Physics, Huazhong University of Science and Technology, Wuhan 430074, China. [2]Dipartimento di Fisica, Politecnico di Milano, Piazza Leonardo da Vinci 32, I-20133 Milano, Italy. [3]IFISC (UIB-CSIC), Instituto de Fisica Interdisciplinar y Sistemas Complejos, E-07122 Palma de Mallorca, Spain. [4]Hubei Key Laboratory of Optical Information and Pattern Recognition, Wuhan Institute of Technology, Wuhan 430205, China. [5]These authors contributed equally: Chengzhi Qin, Han Ye, Shulin Wang. ✉e-mail: wangbing@hust.edu.cn; stefano.longhi@polimi.it; lupeixiang@hust.edu.cn

the interface or potential moves on the lattice with a non-relativistic speed $v$ (at the group velocity scale), the discrete refraction becomes highly dependent on the moving speed[29–33], in contrast to the continuous refraction cases. The reason behind this is that refraction analysis at a moving interface involves a change of reference frame, i.e., Galilean transformation from laboratory to moving frame where Snell's law is applicable. Here Galilean rather than Lorentz transformation is applied considering the non-relativistic moving speed $v \ll c$. One striking effect of space discretization in non-relativistic wave mechanics[34–36] is the breaking of Galilean covariance for discrete Schrödinger equation, making scattering highly dependent on the moving speed[29–33]. By contrast, in the continuous-space limit, Schrödinger equation possesses Galilean invariance, and hence scattering features are not affected by a drift of the interface or potential. Galilean invariance breaking has led to some puzzling scattering phenomena theoretically predicted in recent works. For example, it has been shown that any fast-moving potential becomes reflectionless or even invisible for discretized waves[29–31], whereas the number of bound states sustained by a potential well on a lattice depends on the moving speed owing to a mass renormalization effect[32]. Anderson localization in a moving disordered potential on a lattice also dependents on the drift speed[33], providing another remarkable signature of Galilean invariance breaking in discrete-wave mechanics. From application perspectives, the rich scattering properties enabled by potential moving provide unique discrete-light control strategies going beyond traditional continuous-light schemes of using complementary matching layers[37,38], Kramers-Kronig potentials[39–41], parity-time symmetry[42,43], and transformation optics[44,45], etc. While above studies are based on theoretical analysis, experimental demonstrations on discrete refractions by moving interface or potential remain to date elusive, mainly due to technological difficulties in realizing and controlling moving potentials on a lattice. Additionally, there emerges research interest in pushing discrete-wave mechanics into the optical analog of relativistic regime, where the scalar-wavefunction Schrödinger equation describing single-band dynamics is replaced by the spinor-wavefunction Dirac equation for two-band dynamics, and typical relativistic effects including Klein tunneling[46–48] and *Zitterbewegung*[49,50] have been realized. However, how the relativistic packets are refracted by moving potentials with broken Galilean invariance also remain unexplored both in theory and experiment.

In this work, we theoretically propose and experimentally demonstrate discrete temporal refraction with broken Galilean invariance by a moving gauge-potential barrier in a synthetic lattice created using pulse evolution in two coupled fiber loops[51–56], demonstrating that the scattering features of the barrier are highly velocity-dependent. Firstly, we reveal that due to the two-miniband nature of the lattice, the beam-splitting-free refraction requires the potential to move at a quantized speed of integer $v = 1$ or fractional values $v = 1/q$ (odd $q$) under reflectionless conditions. We also measure the relative barrier-crossing beam delay as the clear signature of single-beam refraction, which manifests asymmetric momentum-dependence feature. Zero beam delay can also be reached for each Bloch momentum via judicious design of gauge-potential difference, showing directional transparency of moving potentials. Symmetric momentum-dependence features can also be attained by using dynamically-modulated moving potentials. Finally, by considering the relativistic limit of light dynamics[46–50], we also observe the refraction of *Zitterbewegung* motion, which exhibits momentum-independent beam delay and omnidirectional transparency condition rooted in the linear band nature of Dirac equation. Our study establishes and demonstrates basic laws for discrete refraction by quantized moving potentials, which may find applications in delay-line designing, precise measurements and signal processing.

## Results

### Discrete temporal refraction and Galilean invariance breaking

We consider temporal beam refraction by a moving potential in a synthetic temporal mesh lattice, which is realized by considering light pulse dynamics in two coupled fiber loops[51–56]. As shown in Fig. 1a, when a single pulse is injected from one loop, it will evolve into a pulse train after successive pulse splitting at central coupler, circulating in two loops and interference at the coupler again. For two loops with lengths $L \pm \Delta L$, the pulse physical time is $t_n^m = mT + n\Delta t$, where $T = L/c_g$ is mean travel time and $\Delta t = \Delta L/c_g \ll T$ is travel-time difference in two loops, $c_g = c/n_g$ is pulse's group velocity, $n_g = 1.5$ is the group index in fiber and $c$ is vacuum light speed. The pulse dynamics can thus be mapped into a "link-node" lattice model $(n, m)$ [Fig. 1c], where $n, m$ denote transverse lattice site and longitudinal evolution step. The leftward/rightward links towards the node correspond to pulse circulations in short/long loops and scattering at each node corresponds to pulse interference at the coupler. Light evolution in the lattice is thus governed by the discretized coupled-mode equations

$$\begin{cases} u_n^{m+1} = [\cos(\beta)u_{n+1}^m + i\,\sin(\beta)v_{n+1}^m]e^{i\phi_u(n-vm)} \\ v_n^{m+1} = [i\,\sin(\beta)u_{n-1}^m + \cos(\beta)v_{n-1}^m]e^{i\phi_v(n-vm)} \end{cases} \quad (1)$$

where $u_n^m, v_n^m$ denote light amplitudes in leftward/rightward links towards node $(n, m)$, $\beta$ is the coupling angle $(0 < \beta < \pi/2)$, corresponding to a splitting ratio of $\cos^2(\beta)/\sin^2(\beta)$ at central coupler. A moving, square-shaped gauge-potential barrier can be created by introducing a non-uniform phase shift distribution (tilted gray ribbon region in Fig. 1c) into the lattice via applying a sliding gate voltage modulation (Fig. 1b) (see also Fig. 2 for detailed experimental realizations)

$$[\phi_u(n - vm), \phi_v(n - vm)] = \begin{cases} (\phi_{u2}, \phi_{v2}), & n_1 \leq n - vm \leq n_2 \\ (\phi_{u1}, \phi_{v1}), & \text{otherwise} \end{cases} \quad (2)$$

where $n - vm = n_{1,2}$ denote two moving boundaries, $n_1, n_2$ are left and right boundary positions at initial step $m = 0$ and $W = n_2 - n_1$ is barrier width. $v = p/q$ is a rational number moving speed, characterizing the barrier moves $p$ sites in every $q$ steps, with $p, q$ being two integers. This rational speed constraint stems from the lattice's double discretization nature both in $n, m$ axes, in contrast to moving potentials in waveguide arrays[29,30], where only transverse axis $n$ is discretized but the longitudinal axis $z$ is still continuously varying. As we shall prove below, further selection rules for the moving speed $v$ will be established based on the intrinsic requirements of discrete refraction. The additional phase shifts $\phi_{u,v}(n-vm)$ are physically associated to an effective scalar $\varphi$ and vector potential $A$[57–59]. Since $\varphi$ and $A$ accumulate phase in time and space[57], $\phi = \int \varphi dt, \phi = \int A dx$, we can relate the phase shifts to $\varphi$ and $A$ through $\phi_{ui} = \int_{m-1}^m \varphi_i dm + \int_{n+1}^n A_i dn = \varphi_i - A_i$, $\phi_{vi} = \int_{m-1}^m \varphi_i dm + \int_{n-1}^n A_i dn = \varphi_i + A_i$, which leads to $\varphi_i \equiv (\phi_{vi} + \phi_{ui})/2$, $A_i \equiv (\phi_{vi} - \phi_{ui})/2$, $i = 1$, 2 refers to the region outside and inside the barrier. It shows that $\varphi$ corresponds to a direct-independent common phase shift in the leftward/rightward links while $A$ corresponds to a direction-dependent phase shift contrast in these two links, both of which are reminiscent of their original physical meanings in electrodynamics[57]. Throughout the paper, we choose a vanishing gauge-potential reference $(\varphi_1, A_1) = (0, 0)$ outside the barrier and $(\varphi_2, A_2) = (\Delta\varphi, \Delta A)$ inside it, where $\Delta\varphi$ and $\Delta A$ are the scalar and vector potential difference.

To analyze the refraction by the moving barrier, we firstly need to choose an appropriate reference frame to apply Snell's law. As shown in Figs. 1c and 1d, there exist two reference frames: the laboratory frame $(n, m)$ where the potential is moving and the moving frame $(n', m')$ where the potential is at rest, which are related to each other through Galilean transformation $n' = n - vm, m' = m$. In the $(n, m)$ frame, since the boundaries are tilted that are not parallel to $m$ axis, Snell's law, i.e., conservation of tangential propagation constants is not

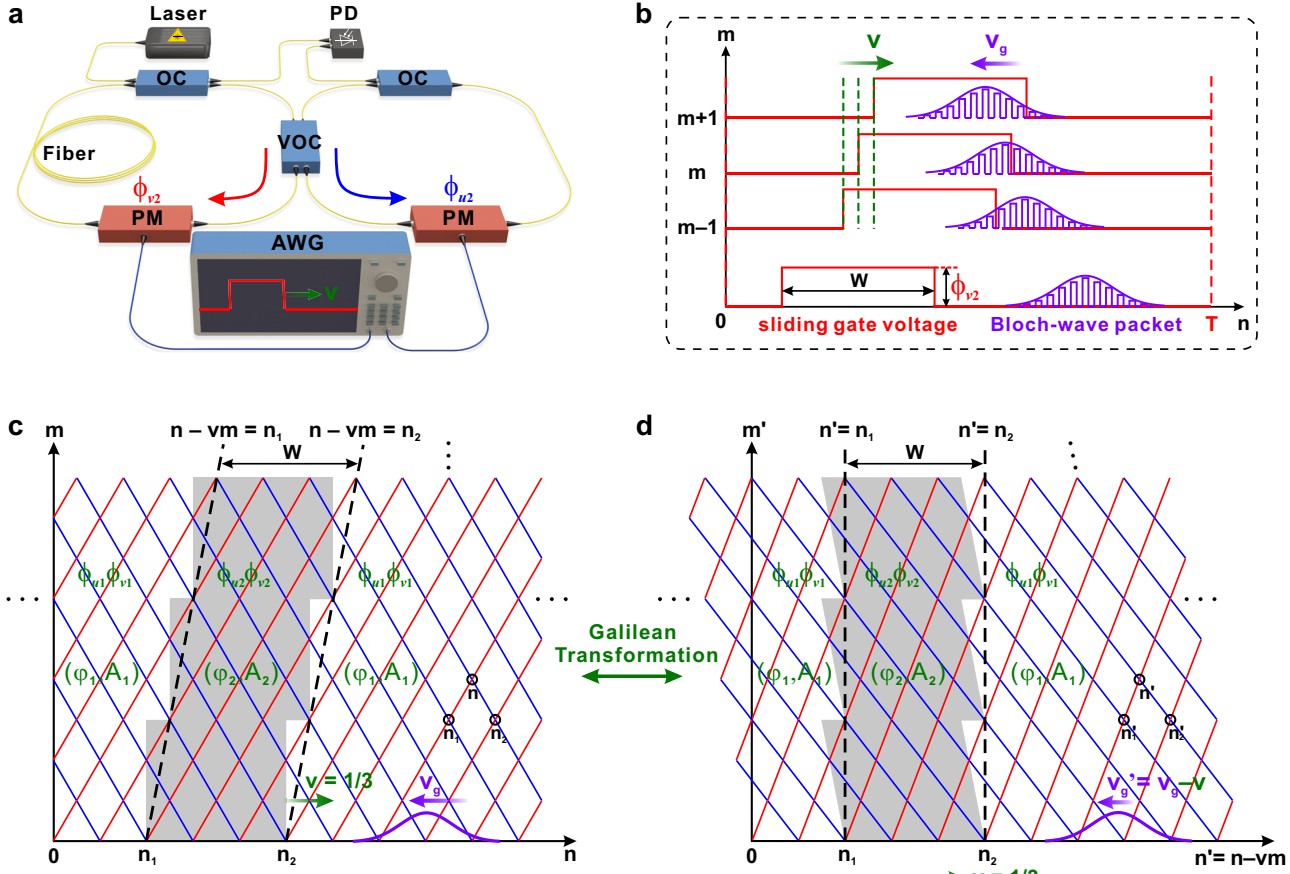

**Fig. 1 | Schematic experimental setup, modulation waveforms, and synthetic lattices. a** Schematic experimental setup of two coupled fiber loops. Blue and red arrows denote light circulation directions in short and long loops. PM: phase modulator; AWG: arbitrary wave generator; PD: photon detector; OC: optical coupler; VOC: variable optical coupler. **b** Schematic modulation waveforms in AWG to generate a moving barrier. The squared gate voltages (red) denote the modulation signals in a period $T$ for initial and three successive steps. $v$ and $v_g$ denote the barrier's moving speed and the packet's group velocity **c** A moving gauge-potential barrier with moving speed $v = 1/3$ in a discrete-time temporal lattice made of leftward (blue) and rightward (red) links towards the node $(n, m)$. The phase modulations are $(\phi_{u2}, \phi_{v2})$ and $(\phi_{u1}, \phi_{v1})$ inside and outside the barrier, corresponding to a gauge-potential distribution of $(\varphi_2, A_2)$ and $(\varphi_1, A_1)$. The two moving boundaries are $n - vm = n_{1,2}$, where $v$ is moving speed, $n_1$, $n_2$ are initial positions of two boundaries and $W = n_2 - n_1$ is barrier width. A Bloch-wave packet (purple) is incident from the right side of barrier. **d** Schematic of the lattice and gauge-potential distribution in the moving frame $(n', m')$ obtained through Galilean transformation from laboratory frame $(n, m)$.

---

applicable. By contrast, in the $(n', m')$ frame, the two boundaries become vertical and Snell's law is applicable. The most unique feature of a moving potential is that its refraction property is highly dependent on the moving speed $v$, owing to Galilean invariance breaking of the underlying equations. To illustrate this point, we consider the coupled-mode equations in the $(n', m')$ frame obtained via Galilean transformation from Eq. (1) [see Supplementary Materials (SM). Section (Sec) 2 for detailed derivation]

$$\begin{cases} f(n',m'+1) = [\cos(\beta)f(n'+1+v,m') + i\,\sin(\beta)g(n'+1+v,m')]e^{i\phi_u(n')} \\ g(n',m'+1) = [i\,\sin(\beta)f(n'-1+v,m') + \cos(\beta)g(n'-1+v,m')]e^{i\phi_v(n')} \end{cases}$$

(3)

where $f(n',m'+1) = u_{n'+vm'}^{m'+1}, g(n',m'+1) = v_{n'+vm'}^{m'+1}$. Note that Eq. (3) depends on potential moving speed $v$ in a nontrivial way, and such a dependence cannot be eliminated by any gauge transformation of wave functions $f$ and $g$. Clearly, the form of Eq. (3) in the moving frame is different from Eq. (1) in the laboratory frame, meaning that the discrete coupled-mode equations are not covariant under Galilean transformation. Therefore, refraction by a moving potential depends on the moving speed $v$, which also differs largely from that of the same

potential at rest ($v = 0$). Breakdown of Galilean invariance stems from the non-parabolic nature of the band structure[29–36,60–62], which can be physically explained in two important limiting cases. In the strong coupling limit $\beta \to \pi/2$, Eq. (1) can be reduced to two decoupled non-relativistic discrete Schrödinger equations[63], where Galilean invariance is breaking owing to space discretization[34–36]. On the other hand, in the weak coupling limit $\beta \to 0$, Eq. (1) at the long-wavelength limit reduces to a relativistic Dirac equation [see Eq. (12) below][46–50], which is clearly invariant for a Lorentz (rather than a Galilean) boost.

The refraction should be analyzed using band structure matching approach. In the $(n, m)$ frame, the eigen Floquet-Bloch mode in each uniform region $i = 1, 2$ is $(u_n^m, v_n^m)^T = (U, V)^T \exp(ikn) \exp(-i\theta m)$, where $\theta = \theta_\pm^{(l)}(k) = \pm\cos^{-1}[\cos(\beta)\cos(k - A)] - \varphi + 2\pi l$ is Floquet band structure, $k$, $\theta_\pm^{(l)}(k)$ are transverse Bloch momentum and longitudinal propagation constant of $l$-th Floquet band, $l = 0, \pm 1, \pm 2, ...$ is Floquet order, "$\pm$" denote positive and negative minibands. The physical effect of scalar and vector potential is thus to induce Bloch momentum and propagation constant shift. Applying Galilean transformation $(u_n^m, v_n^m)^T = (U, V)^T \exp[ik(n'+vm')]\exp(-i\theta m') = (U, V)^T\exp(ikn')\exp[-i(\theta-vk)m']$ and comparing with the eigen mode $(U', V')^T\exp(ik'n')\exp(-i\theta'm')$ in the $(n', m')$ frame, we can obtain the Floquet band structure in the $(n', m')$

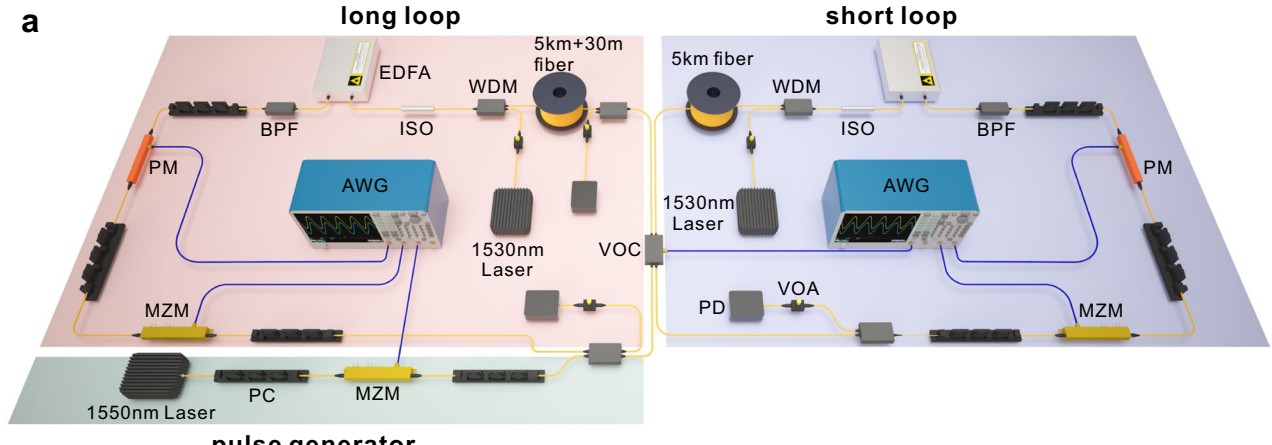

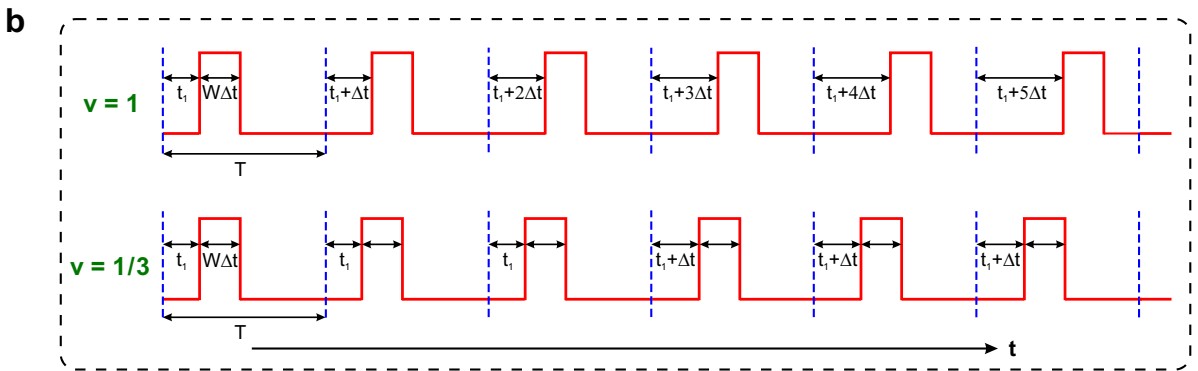

**Fig. 2 | Layout of experimental setup and schematic modulation waveforms to generate a moving gauge-potential barrier. a** All optical and electrical components are as follows: Polarization controller (PC); Mach-Zehnder modulator (MZM); Optical coupler (OC) with a constant splitting ratio and variable optical coupler (VOC) with a programable splitting ratio; Arbitrary waveform generator (AWG); Single mode fiber (SMF); Photodiode (PD); Variable optical attenuator (VOA); Wavelength division multiplexer (WDM); Erbium-doped fiber amplifier (EDFA); Band-pass filter (BPF) and phase modulator (PM). **b** Upper and lower waveforms show the sliding gate voltages in long loop to generate a moving gauge-potential barrier with moving speeds $v = 1$ and $v = 1/3$. $\Delta t$, $T$ correspond to the time interval between adjacent lattice sites and modulation period, and $t_1$ denotes the timing of rising edge of gate voltage in the first period. The barrier width is $W$, corresponding to a duration time of $W\Delta t$ for the gate voltage signal.

frame

$$\theta'^{(l)}_{\pm}(k') = \theta^{(l)}_{\pm}(k) - vk$$
$$= \pm\cos^{-1}[\cos(\beta)\cos(k - A)] - \varphi - vk + 2\pi l, \quad (4)$$

and $k' = k$, $(U',V')^T = (U, V)^T$. The Galilean transformation does not change the Bloch momentum and eigenstate but can modify the band structure. Specifically, the band structure in the $(n', m')$ frame acquires a ramped term $-vk$, i.e., a tilt, compared to that in the $(n, m)$ frame, making the band structure matching also $v$ dependent for refraction analysis. The group velocity in $(n', m')$ frame is $v'_{g,\pm}(k) = \partial\theta'^{(l)}_{\pm}(k)/\partial k = v_{g,\pm}(k) - v$, indicating that the packet acquires an additional velocity term $-v$ in the moving frame, which is a direct consequence of Galilean transformation. Since the refraction is $v$-dependent due to Galilean invariance breaking, below we will identify the selection rules for $v$ to enable a well-defined discrete refraction.

### Selection rules of potential moving speed $v$ for a well-defined refraction

Now we identify the selection rules for the potential moving speed $v$ to ensure a well-defined refraction. Such selection rules basically arise from multiband nature of the Floquet synthetic lattice. For a generic $v$, refraction always exists but reflection may vanish[29,30]. Meanwhile, due to the multiple Floquet bands and two minibands nature of the lattice, an incident packet will generally split into multiple refracted beams. To ensure a well-defined refraction, such beam splitting should be eliminated. Below, we shall clarify the required condition of $v$ to eliminate both reflection and beam splitting. Let us take the first refraction at right boundary as an example, which is also applicable to second refraction at left boundary. In our analysis, we only consider propagative waves with real number Bloch momenta. The more rigorous analysis involving evanescent waves is outlined in SM. Sec 2. Note that the evanescent waves decay to zero away from the interface, making no observable contributions to the refraction. Consider a Bloch-wave packet at $(k_i, \theta'_i) = (k_{1,+}, \theta'^{(0)}_{1,+})$ in "+" miniband incident from right side of the barrier (blue circles in Fig. 3a). $\beta_1 = \beta_2 = \beta$ is chosen in the two regions. For a large integer speed $v = p$ or fractional $v = p/q$ satisfying $v > |v_{g,\pm}(k)|_{max} = \cos(\beta)$, Floquet bands are tilted down monotonically with $v'_{g,\pm}(k) = v_{g,\pm}(k) - v < 0$ for every $k$. Since the barrier moves faster than group velocity upper bound, beam reflection is forbidden for any incident $\theta'_i$. This case is shown in Fig. 3a and Fig. S1 for $v = 1$, $\beta = \pi/3$. Meanwhile, only one refracted packet is matched at $(k^{(l)}_{2,+}, \theta'^{(l)}_{2,+})$ or $(k^{(l)}_{2,-}, \theta'^{(l)}_{2,-})$ for "+" or "−" miniband in $l$-th Floquet order (red circles in Fig. 3a). On the contrary, for a slow, fractional moving speed $v = p/q < |v_{g,\pm}(k)|_{max} = \cos(\beta)$, reflection is eliminated only in specific ranges of $\theta'_i$ (white regions outside gray ribbons in Fig. 4a and Fig. S2 for $v = 1/3$, $\beta = \pi/3$). While in other ranges of $\theta'_i$ (gray ribbons), reflection occurs at specific $k$ with $v'_{g,\pm}(k) > 0$ (black circles in Fig. 4a). Meanwhile, multiple refracted packets can be matched for each miniband, which are spaced in $k$ other than $2\pi$ and possess different $v_g$ to cause beam splitting. The reflective ranges can be further tuned by $\beta$, varying from full to partial reflectionless, ultimately to full reflective

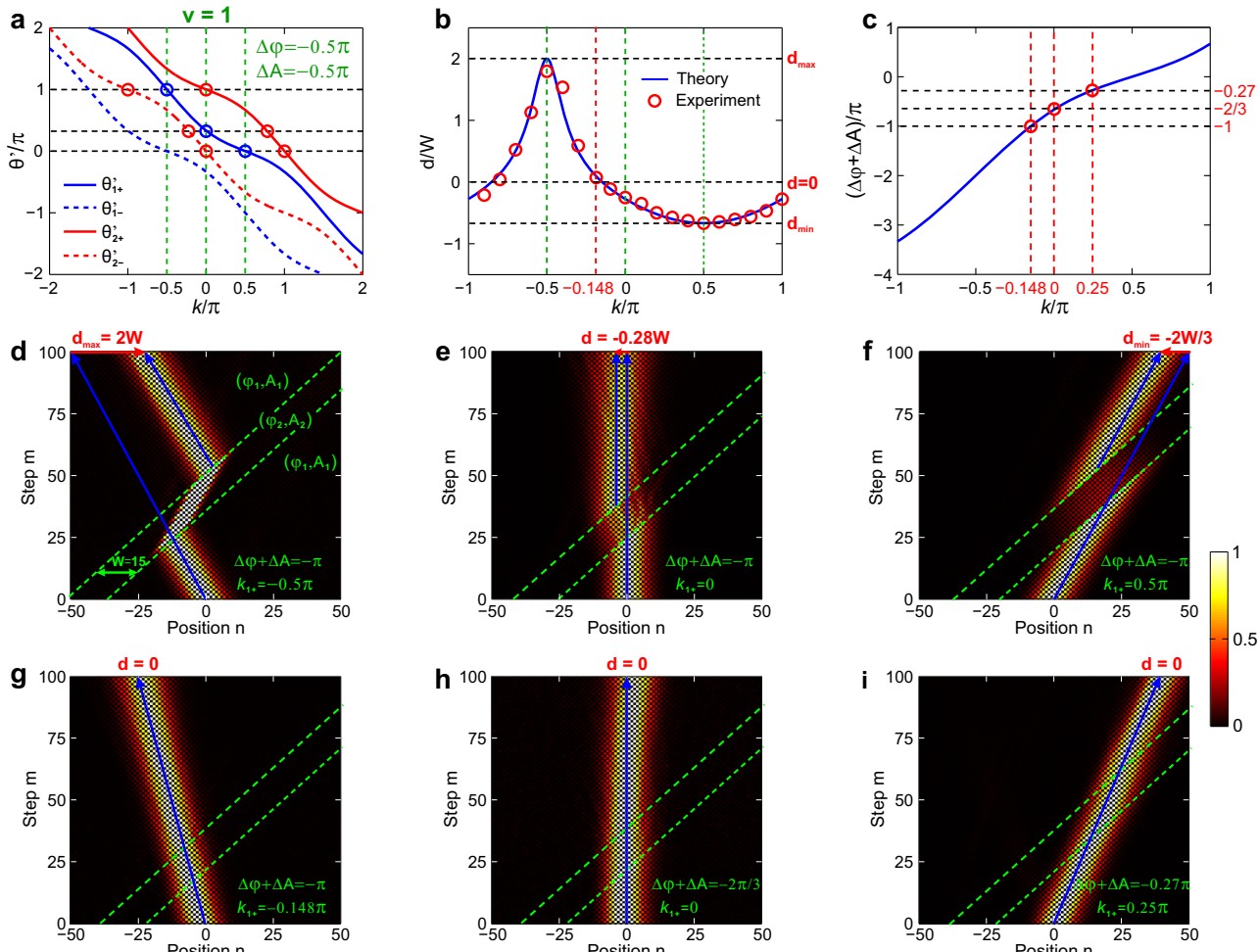

**Fig. 3 | Band structures, measured beam delay and refractions for integer moving speed $v = 1$. a** Floquet bands of $l = 0$ for positive (solid) and negative (dashed) minibands outside (blue) and inside (red) the barrier with scalar and vector potential distribution $(\varphi_1, A_1) = (0, 0)$, $(\Delta\varphi, \Delta A) = (-\pi/2, -\pi/2)$, $\beta = \pi/3$, and $W = 15$. The incident, transmitted and refracted packets are denoted by blue and red circles. **b** Calculated (blue curve) and measured (red circles) beam delays versus incident Bloch momentum $k_{1+}$, which reaches $d_{max} = 2 W$, $d_{min} = -2 W/3$ and $d = 0$ at $k_{1+} = -0.5\pi$, $0.5\pi$, and $-0.148\pi$. **c** Required gauge-potential difference combination $\Delta\varphi + v\Delta A$ to achieve a transparency condition for each targeted $k_{1+}$. **d**–**f** Measured field evolutions for $k_{1+} = -0.5\pi$, 0 and $0.5\pi$, corresponding to $d_{max} = 2 W$ (**d**), $d = -0.28 W$ (**e**) and $d_{min} = -2 W/3$ (**f**). **g**–**i** Measured field evolutions for $k_{1+} = -0.148\pi$, 0 and $0.25\pi$ under $\Delta\varphi + v\Delta A = -\pi$, $-2\pi/3$, and $-0.27\pi$, corresponding to a transparent moving potential for a leftward (**g**), frozen (**h**), and rightward (**i**) propagating packet. In (**d**–**i**), the green dashed lines denote the two moving boundaries of moving potential. The blue solid lines denote the incident and transmitted packets' trajectories in the $(n, m)$ plane.

as $\beta$ decreases (see Fig. S3). For a given $\beta$, the full reflective regime can be always reached by choosing a sufficiently slow moving speed $v \to 0$.

Under the reflectionless condition, let's reveal the selection rules for $v$ to eliminate beam splitting from "±" minibands and different Floquet orders. By applying Snell's law $\theta_{2,\pm}^{\prime(l)}(k_{2,\pm}^{(l)}) = \theta_i'$, we can obtain

$$\pm \cos^{-1}[\cos(\beta)\cos(k_{2,\pm}^{(l)} - \Delta A)] - \Delta\varphi - vk_{2,\pm}^{(l)} + 2\pi l = \theta_i', \quad (5)$$

No beam splitting requires group velocity degeneracy for "±" minibands: $v_{g,+}(k_{2,+}^{(l)}) = v_{g,-}(k_{2,-}^{(l)})$, which further requires $\sin(k_{2,+}^{(l)}) = -\sin(k_{2,-}^{(l)})$. Since $k_{2,+}^{(l)} - k_{2,-}^{(l)} < 2q\pi$ in the same $l$-th Floquet band, they can only be spaced by $k_{2,+}^{(l)} - k_{2,-}^{(l)} = q\pi$, $q = 1, 3, 5, \ldots$ is an odd number. Equation (5) can be thus rewritten as

$$\begin{cases} \cos(\theta_i' - 2\pi l + vk_{2,-}^{(l)} + \Delta\varphi + vq\pi) = \cos(\beta)\cos(k_{2,-}^{(l)} - \Delta A + q\pi) \\ \cos(\theta_i' - 2\pi l + vk_{2,-}^{(l)} + \Delta\varphi) = \cos(\beta)\cos(k_{2,-}^{(l)} - \Delta A) \end{cases}, \quad (6)$$

which leads to $vq\pi = q'\pi$, $q' = 1, 3, 5 \ldots$ is also an odd number. Since $k_{2,+}^{(l)}, k_{2,-}^{(l)}$ are two closest solutions of Eq. (6) in $l$-th Floquet order, $q' = 1$

should be chosen, which yields the quantization condition for $v$

$$v = \frac{1}{q} = \begin{cases} 1, & \text{(integer)} \\ 1/3, 1/5, \ldots, & \text{(fractional)} \end{cases}, \quad (7)$$

The speed selection rule stems from two-miniband nature of the lattice, which doesn't exist in single-band waveguide arrays[29,30]. Under this condition, the refracted packets in adjacent Floquet orders satisfy $k_{2,\pm}^{(l+1)} - k_{2,\pm}^{(l)} = 2\pi/v = 2q\pi$, meaning that they also share the same $v_g$ to cause no beam splitting. On the contrary, for an unpermitted integer speed $v = p \neq 1$ or fractional one $v = p/q_1 \neq 1/q$ (odd $q$), we can get $k_{2,\pm}^{(l+p)} - k_{2,\pm}^{(l)} = 2\pi p/v = 2q_1\pi$, indicating the refracted packets with Floquet indices spaced by $p$ share the same $v_g$, leading to $2p$ splitting beams during refraction. Below, we will choose two permitted speed of integer $v = 1$ and fractional $v = 1/3$ for experimental verifications. The simulation results for other counterexamples with unpermitted integer speeds $v = 2, 3$ and fractional values $v = 1/2, 3/2$ are discussed in SM (Figs. S4 and S5).

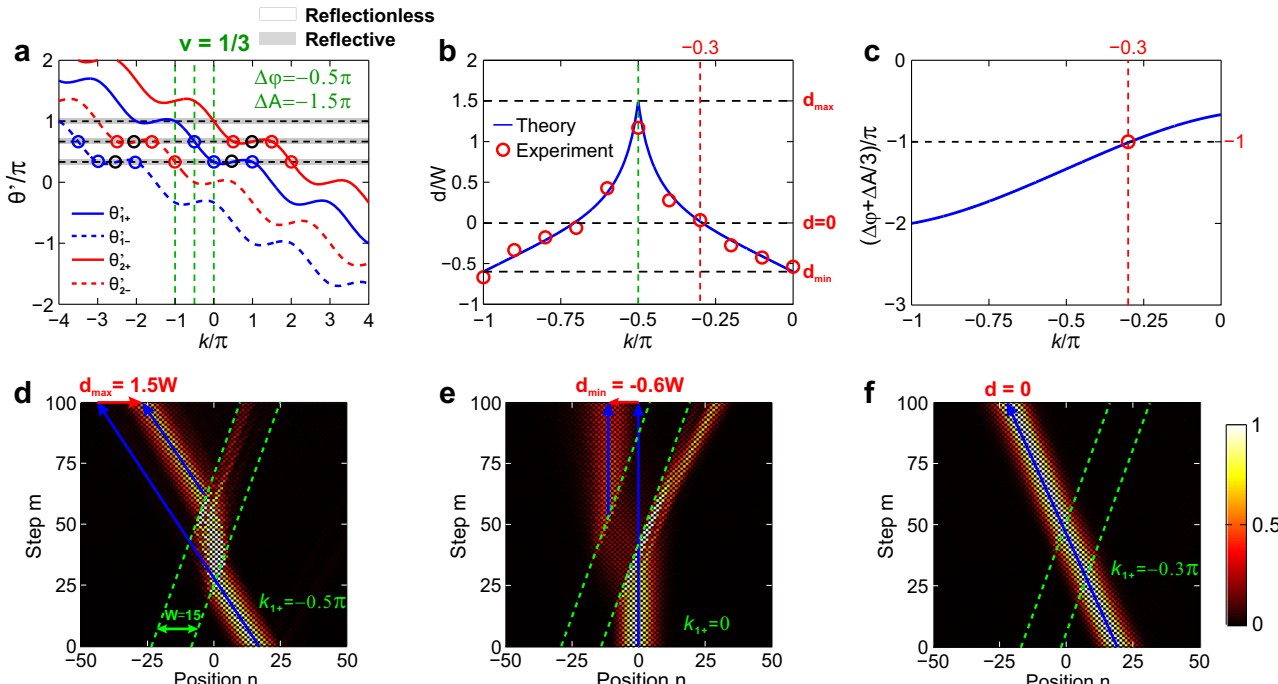

**Fig. 4 | Band structures, measured beam delay and refractions for fractional moving speed $v = 1/3$. a** Floquet bands $l = 0$ for positive (solid) and negative (dashed) minibands outside (blue) and inside (red) the barrier with $(\varphi_1, A_1) = (0, 0)$, $(\Delta\varphi, \Delta A) = (-\pi/2, -3\pi/2)$, $\beta = \pi/3$, and $W = 15$. The incident, transmitted, refracted, and reflected packets are denoted by blue, red, and black circles. The gray ribbons denote the reflective regions. **b** Calculated (blue curve) and measured (red circles) beam delays versus incident Bloch momentum, which reaches $d_{max} = 1.5\,W$,

$d_{min} = -0.6\,W$, and $d = 0$ at $k_{1+} = -0.5\pi$, 0, and $-0.3\pi$ (or $-0.7\pi$). **c** Required gauge-potential difference combination $\Delta\varphi + \Delta A/3$ to reach transparency condition for each Bloch momentum. **d**–**f** Measured field evolutions for $k_{1+} = -0.5\pi$, 0, and $-0.3\pi$ under $\Delta\varphi + \Delta A/3 = -\pi$, corresponding to above three cases of $d_{max} = 1.5\,W$ (**d**), $d_{min} = -0.6\,W$ (**e**), and $d = 0$ (**f**). In (**d**–**f**), the green dashed lines denote the two moving boundaries of moving potential. The blue solid lines denote the incident and transmitted packets' trajectories in the $(n, m)$ plane.

## Refraction from a potential barrier with quantized moving speed: experimental results

In this section, we present experimental demonstrations of velocity-dependent temporal beam refraction using two permitted moving speeds of integer $v = 1$ and fractional $v = 1/3$. In the experiment, the observation of a single transmitted wave packet is the clear signature of beam-splitting elimination. The refraction process can be quantitatively characterized by a relative beam delay, denoting the packet's transverse propagation difference with and without the barrier. The packet's transit time in the barrier is $\tau = W/|v'_{g,+}(k_{2,+})| = W/|\cos(\beta) \sin(k_{2,+} - \Delta A) - v|$, corresponding to a relative beam delay

$$d(k_{1,+}) = [v_{g,+}(k_{2,+}) - v_{g,+}(k_{1,+})]\tau$$

$$= \frac{W \cos(\beta)[\sin(k_{2,+} - \Delta A) - \sin(k_{1,+})]}{|v - \cos(\beta)\sin(k_{2,+} - \Delta A)|}. \quad (8)$$

Hereafter we will omit Floquet index $l$ by choosing $l = 0$. $d > 0$ ($d < 0$) denotes the cases of beam delay and advance. Since $d$ can be tuned continuously from positive to negative values by varying $k_{1,+}$, there must exist a specific $k_{1,+}$ where $d = 0$ is reached. This case corresponds to a transparency condition, where both refraction and reflection are eliminated. $d = 0$ requires the group velocity degeneracy inside and outside the barrier, $v_{g,+}(k_{2,+}) = v_{g,-}(k_{2,-}) = v_{g,+}(k_{1,+})$, which can only be fulfilled provided that $k_{2,-} - \Delta A = -k_{1,+}$, $k_{2,+} - \Delta A = \pi - k_{1,+}$. By combining with Eq. (8), we can obtain the transparency condition

$$\Delta\varphi + v\Delta A = 2vk_{1,+} - 2\cos^{-1}[\cos(\beta)\cos(k_{1,+})]. \quad (9)$$

therefore, we can achieve a directional transparent moving potential for any targeted incident $k_{1,+}$ by designing an appropriate gauge-potential combination of $\Delta\varphi + v\Delta A$.

Both the beam delay and transparency condition have been verified by our refraction experiments. The experiment setup is shown in Fig. 2a, which consists of two fiber loops with lengths of $L \pm \Delta L \sim 5\,km \pm 15\,m$, corresponding to $T \sim 25\,\mu s$ and $\Delta t \sim 75\,ns$. The initial pulse is generated from a 1550 nm distributed-feedback continuous laser, which is cut into a ~50 ns duration pulse by a Mach-Zehnder modulator (MZM) and injected from the long loop. To record the pulse intensity evolution, we couple a portion of light from both loops and detect them using photodiodes (PDs) after each circulation step. The phase shifts are applied through phase modulators (PMs) driven by the arbitrary wave generators (AWGs) with programmable modulation signals. The required moving speed $v$ is attained by precisely controlling the relative delay of the sliding gate voltage between adjacent modulation periods (Fig. 2b). More details about experimental setup and measurement techniques are discussed in "Methods".

Firstly, we choose integer moving speed $v = 1$, $\beta = \pi/3$, $W = 15$, $\Delta\varphi + v\Delta A = -\pi$ using $(\Delta\varphi, \Delta A) = (-\pi/2, -\pi/2)$ by applying $(\phi_{u2}, \phi_{v2}) = (\varphi_2 - A_2, \varphi_2 + A_2) = (0, -\pi)$. In this case, we only need to introduce a $\pi$-phase shift into long loop to simplify the experiment. Figure 3a shows "+" (solid) and "−" (dashed) minibands outside (blue) and inside (red) the barrier for $l = 0$ Floquet band. The measured beam delay $d$ versus $k_{1,+}$ is shown in Fig. 3b, which agrees fairly with theoretical curve in Eq. (8). Note that $d$ is asymmetric for $\pm k_{1,+}$, which reaches $d_{max} = 2\,W\cos(\beta)/|v-\cos(\beta)| = 2W = 30$, $d = 0.28\,W = -4.2$ and $d_{min} = -2W\cos(\beta)/|v+\cos(\beta)| = -2W/3 = -10$ for $k_{1,+} = -\pi/2$, 0 and $\pi/2$, as shown in Figs. 3d-3f. Figure 3c shows the required gauge-potential combination $\Delta\varphi + v\Delta A$ to realize the transparency condition for each $k_{1,+}$. Three representative cases are shown in Figs. 3g-3i, where a leftward ($k_{1,+} = -0.148\pi$), frozen ($k_{1,+} = 0$) and rightward ($k_{1,+} = 0.25\pi$) propagating packets become refractionless and reflectionless for $\Delta\varphi + v\Delta A = -\pi, -2\pi/3$ and $-0.27\pi$, thus verifying the theoretical analysis in Eq. (9).

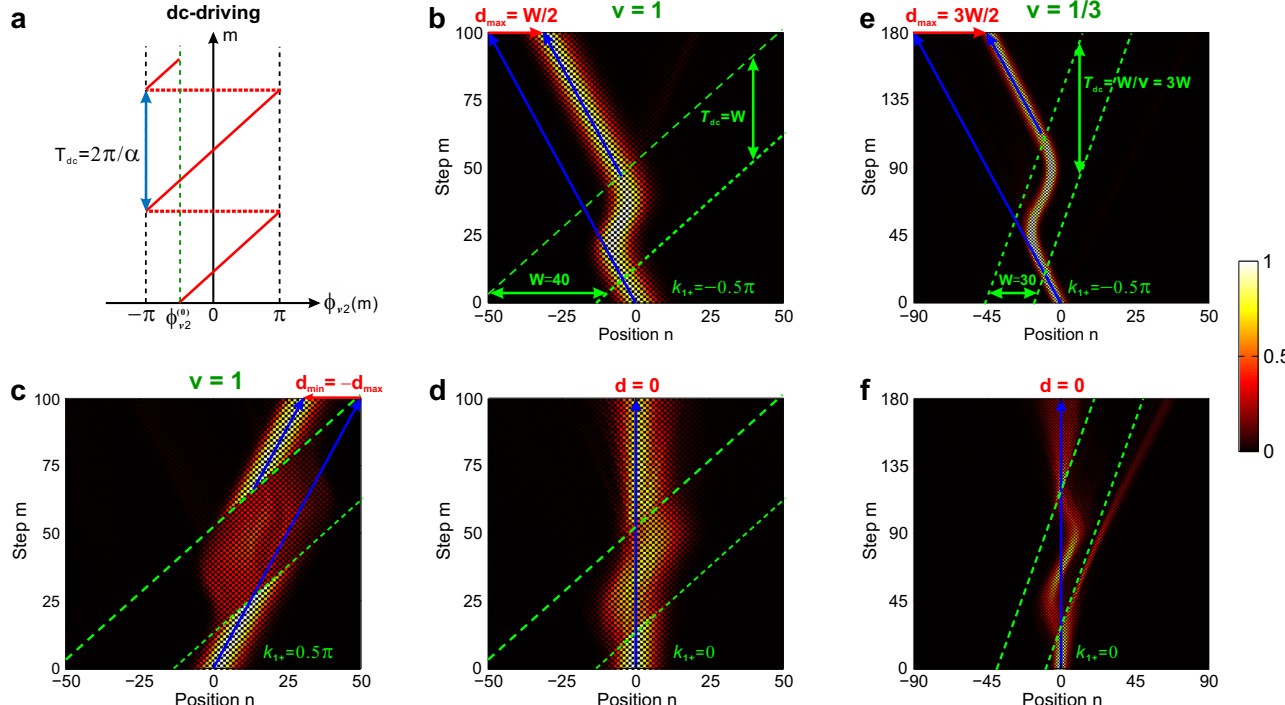

**Fig. 5 | Measured beam refractions by a dc-driving, moving gauge-potential barrier. a** Schematic waveform of dc-driving phase modulation. $T_{dc} = 2\pi/\alpha$ is dc-driving period, $\alpha$ is dc-driving force and $\phi_{v2}(0)$ is the initial phase in the long loop. **b–d** Measured field evolutions for $k_{1+} = -0.5\pi$, $0.5\pi$, and 0, corresponding to rightward $d_{max} = W/2$ (**b**), leftward $d_{min} = -W/2$ (**c**), and $d = 0$ (**d**) under $v = 1$, $(\Delta\varphi, \Delta A) = (-\pi/2, -\pi/2)$, $A_2(0) = -0.5\pi$, $\beta = \pi/3$ and $T_{dc} = W = 40$. **e, f** Measured field

evolutions for $k_{1+} = -0.5\pi$, 0, corresponding to rightward $d_{max} = 3W/2$ (**e**) and $d = 0$ (**f**) under $v = 1/3$, $(\Delta\varphi, \Delta A) = (-\pi/2, -3\pi/2)$, $A_2(0) = -0.25\pi$, $\beta = \pi/3$, and $T_{dc} = W/v = 90$ for $W = 30$. In (**b–f**), the green dashed lines denote the two moving boundaries of the moving potential. The blue solid lines denote the incident and transmitted packets' trajectories in the $(n, m)$ plane.

Then, we turn to the fractional moving speed $v = 1/3$. Figure 4a shows the $l = 0$ Floquet band and Fig. 4b shows measured $d$ versus $k_{1+}$, where $\Delta\varphi + \Delta A/3 = -\pi$ for $(\phi_2, A_2) = (-\pi/2, -3\pi/2)$ by applying $(\phi_{u2}, \phi_{v2}) = (-\pi, 0)$, such that we only need to introduce a $\pi$-phase shift into short loop. We achieve $d_{max} = W\cos(\beta)/v = 3W/2$ and $d_{min} = -W\cos(\beta)/|v + \cos(\beta)| = -3W/5$ at $k_{1+} = -\pi/2$ and 0 in Fig. 4d, e. Since $v = 1/3 < \cos(\beta) = 1/2$, there truly emerge beam reflections at left boundary for $k_{1+} = -\pi/2$ and at right boundary for $k_{1+} = 0$, verifying above analysis about reflectionless condition. Although $k_{1+} = -\pi/2$ and 0 enter the reflective range (gray ribbons), the two refracted packets happen to share the same $v_g$ to show no beam splitting (see Fig. S2 for detailed analysis). Figure 4c shows the required $\Delta\varphi + \Delta A/3$ to achieve transparency condition, which is also evidenced by an example with $k_{1+} = -\pi/3$, $\Delta\varphi + \Delta A/3 = -0.3\pi$ shown in Fig. 4f.

**Refraction by a dynamically-modulated, moving gauge-potential barrier**
In previous sections, we consider a sliding potential where $(\varphi_2, A_2)$ doesn't vary with evolution step $m$ while moving. Here we apply an additional dynamic modulation $[\varphi_2(m), A_2(m)]$ to the moving potential to modify its refraction properties. A simultaneous spatially-distributed and time-modulated potential is referred to as space-time potential, which can usually possess distinct scattering properties beyond the static counterpart arising from Galilean invariance violation solely.

As illustrative examples, we choose two typical dc- and ac-driving cases with linearly-varying and periodically-oscillating vector potential $A_2(m) = A_2(0) + \alpha m$ and $A_2(m) = A_m\cos(\omega m + \varphi_m)$, where $\alpha = 2\pi/T_{dc}$ is dc-driving force and $A_2(0)$ is the initial phase. $A_m$, $\omega = 2\pi/T_{ac}$ and $\varphi_m$ are ac-driving amplitude, frequency, and initial phase. The dc- or ac-driving $A_2(m)$ corresponds to a constant or a time-oscillating electric field, which can induce Bloch oscillations (BOs) or directional

transport. The refracted Bloch momenta thus evolve as $k_{2,\pm}(m) = k_{2,\pm}(0) - A_2(m)$, corresponding to the instantaneous group velocities $v_{g,\pm}[k_{2,\pm}(m)] = \pm\cos(\beta)\sin[k_{2,\pm}(0) - A_2(m)]$, where $k_{2,\pm}(0)$ are initial Bloch momenta. When we choose the permitted $v$ in Eq. (7), the "±" mini-bands always share the same $v_g$ as $m$ varies and beam splitting is still eliminated as in the unmodulated case.

Usually, for a slow driving frequency, i.e., with small $\alpha$ or $\omega$, the adiabaticity and continuous-time approximation are valid, we can define a $m$-independent averaging group velocity in one driving period

$$\langle v_{g,\pm}(k_{2,\pm})\rangle = \frac{1}{T}\int_0^T v_{g,\pm}[k_{2,\pm}(m)]dm$$
$$= \begin{cases} 0, & (dc) \\ \pm J_0(A_m)\cos(\beta)\sin[k_{2,\pm}(0)], & (ac) \end{cases} \quad (10)$$

where $J_0$ is 0-th Bessel function. For ac-driving case, we are interested in the dynamic localization (DL) effect[54] occurring as $J_0(A_m) = 0$, which leads to $\langle v_{g,\pm}(k_{2,\pm})\rangle = 0$. Meanwhile, to guarantee a well-defined beam delay, the barrier's transit time should be an integer multiple of the driving period, $\tau = W/v = sT_{dc} = sT_{ac}$, where $s$ is an integer. Under BOs or DL, the beam delay can be uniformly written as

$$d(k_{1,+}) = [\langle v_{g,+}(k_{2,+})\rangle - v_{g,+}(k_{1,+})]\tau = \frac{-W\cos(\beta)\sin(k_{1,+})}{v}. \quad (11)$$

Unlike above static moving potential where $d$ is asymmetric for $\pm k_{1,+}$, the dynamically-modulated case enables symmetric $d$ for $\pm k_{1,+}$. This symmetric momentum-dependent feature is due to periodic nature of BOs (or DL) and hence no net transport in the barrier. The transparency condition is achieved at $k_{1,+} = 0$ and $v$-independent, which is also different from the static case in Eq. (9).

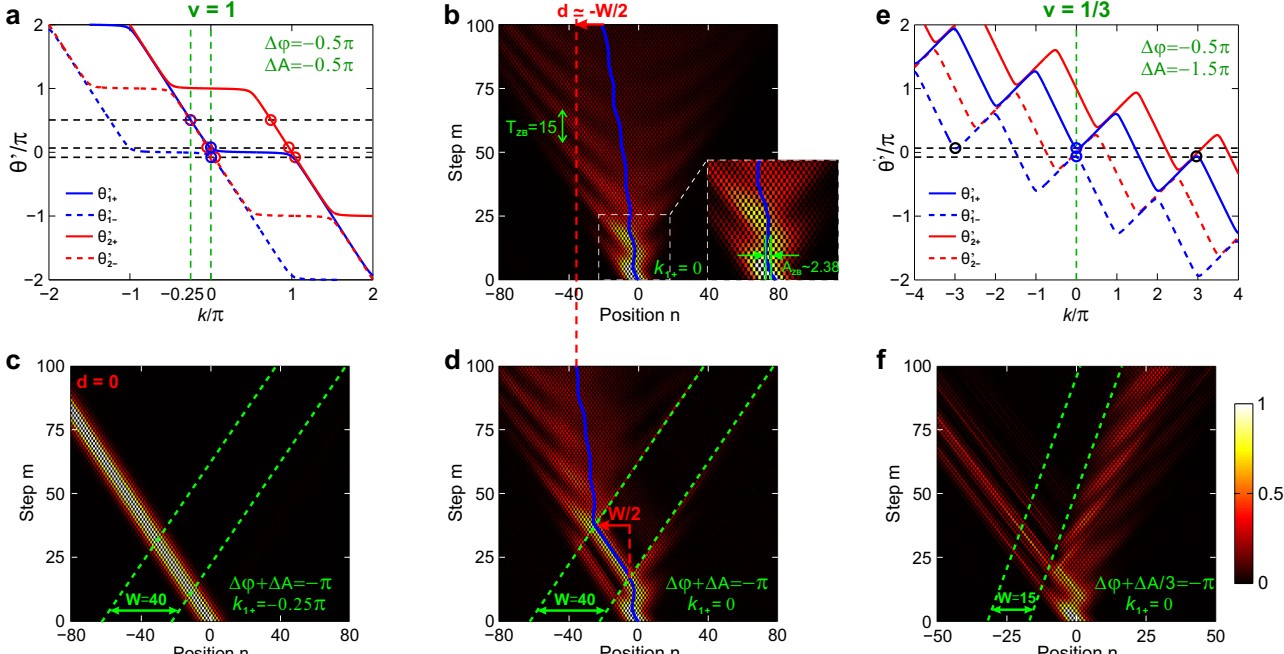

**Fig. 6 | Band structures and measured beam refractions for the relativistic packet displaying ZB. a** Floquet band $l = 0$ for $v = 1$, $\beta = \pi/15$ and $(\Delta\varphi, \Delta A) = (-\pi/2, -\pi/2)$. The incident, transmitted and refracted packets are denoted by blue and red circles. **b** Measured field evolution for ZB without the moving barrier for an incident packet with $k_{1+} = 0$ and $(U_0, V_0) = (1, 0)$. **c, d** Measured field evolutions in the presence of moving barrier excited at $k_{1+} = -\pi/4$ (**c**) and 0 (**d**). The blue curves in (**b**) and (**d**) denote the extracted experimental center-of-mass (COM) trajectories of ZB.

The beam delay is $d = -W/2$. In (**b–d**), the barrier width is $W = 40$. **e** Floquet band $l = 0$ for $v = 1/3$, $\beta = \pi/15$ and $(\Delta\varphi, \Delta A) = (-\pi/2, -3\pi/2)$. **f** Measured field evolution for $k_{1+} = 0$, $(U_0, V_0) = (1, 0)$ and $W = 15$, showing large reflection at first boundary of the moving barrier. In (**b–d, f**), the green dashed lines denote the two moving boundaries of the moving potential. The blue solid lines denote the incident and transmitted packets' trajectories in the $(n, m)$ plane.

The experimental results for dc-driving case are shown in Fig. 5 and ac-driving cases are presented in SM. Sec. 3. Figures 5b-5d illustrate integer speed $v = 1$ and $\tau = W = T_{dc}$, clearly showing symmetric $d = \pm W/2$ for $k_{1+} = -\pi/2, \pi/2$ and $d = 0$ for $k_{1+} = 0$. While for fractional speed $v = 1/3$, $\tau = 3W = T_{dc}$ in Figs. 5e, f, we obtain $d_{max} = 3W/2$ for $k_{1+} = -\pi/2$ and $d = 0$ for $k_{1+} = 0$, verifying the transparency condition. Note that at $k_{1+} = -\pi/2$, reflection vanishes at left boundary, different from the static case in Fig. 4d. The reason is that for $\theta_i'$ initially locating in the reflective range, the driving shifts it out of the reflective regime to eliminate beam reflection. The elimination of reflection by dynamically-modulated moving potential expands our control capabilities over refractions, which is firstly realized in this work.

### Light dynamics in the relativistic regime: the refraction of *Zitterbewegung* motion

Finally, we shall push the wave dynamics into optical analog of relativistic limit and study the unique refraction features in this regime[46–50]. To this end, we choose the weak coupling limit $\beta \to 0$, where we can get a very narrow band gap $\Delta_g = |\theta_+(0) - \theta_-(0)| = 2\beta$ between "$\pm$" minibands. The two bands also become linear in the whole Brillouin zone, $\theta_{\pm}'(k) = \pm\cos^{-1}[\cos(\beta)\cos(k)] - vk = (\pm 1 - v)k$, possessing two constant group velocities of $v_{g,\pm}'(k) \equiv \pm 1 - v$ and nearly touch at $k = 0$. Accordingly, a packet excited at $k = 0$ will manifest relativistic *Zitterbewegung* (ZB)[49,50], an oscillatory trembling motion due to the interference (beat) between "$\pm$" minibands. How does the refraction behave in the relativistic limit by the moving potential is an interesting question, because in the continuous (long-wavelength) limit $k \to 0$, Galilean invariance is still broken since the two-miniband lattice model described by Eq. (1) reduces to a Dirac (see below) rather than two decoupled Schrödinger equations, which is covariant for Lorentz rather than Galilean boost.

In the absence of moving barrier, wave dynamics near $k = 0$ is described by the Dirac-type equation, derived from Eq. (1) under $\beta \to 0$ (see SM. Sec. 4 for this derivation)

$$i\frac{\partial\psi}{\partial m} = i\sigma_z\frac{\partial\psi}{\partial n} - \beta\sigma_x\psi, \tag{12}$$

where $\sigma_x$, $\sigma_z$ are Pauli matrices, $\psi(n, m) = [U(n, m), V(n, m)]^T$ is the spinor wave function. For a packet excited at $k = 0$, $\psi(n, 0) = (1, 0)^T\exp[-n^2/(w_0)^2]$, the center-of-mass (COM) trajectory of ZB is calculated as $\langle n_{COM}(m)\rangle = \langle n_{COM}(0)\rangle + v_0 m + A_{ZB}\sin(\omega_{ZB}m + \varphi_{ZB})$ (see SM. Sec. 4 for detailed derivation), consisting of a linear drift and a sinusoidally-oscillating term, where $\langle n_{COM}(0)\rangle$ is the input position, $v_0 = -1/(\beta w_0)^2$, $A_{ZB} = 1/(2\beta)$, $\omega_{ZB} = 2\beta = \Delta_g$ ($T_{ZB} = \pi/\beta$) and $\varphi_{ZB} = \pi$ are mean drift velocity, ZB's oscillating amplitude, frequency (period) and initial phase, $w_0$ is the input packet's width.

In the presence of moving barrier, the input packet displaying ZB at $k = 0$ (denoted by blue circles) can match two refracted packets (red circles) locating in the linear regime $k \neq 0$ far away from the band gap (Fig. 6a). As a result, ZB comes to a complete halt in the barrier and turns into directional transport with a constant group velocity $v_{g,+}(k_{2,+}) \equiv -1$. After crossing the barrier, ZB is restored. The transit time is $\tau = W/|v_{g,+}'(k_{2,+})| \equiv W/(1+v)$, corresponding to a momentum-independent relative beam delay

$$d = [v_{g,+}(k_{2,+}) - v_0]\tau \equiv -W\left(\frac{1+v_0}{1+v}\right). \tag{13}$$

Usually for a broad packet, $|v_0| = 1/(\beta w_0)^2 \ll 1$ can be neglected, we get $d = -W/(1+v)$. The beam delay of ZB is thus independent of gauge-

potential difference $\Delta\varphi$ or $\Delta A$. This is due to the linear band nature of Dirac equation, which is in stark contrast to previous non-relativistic cases.

In experiment, we firstly choose the case without the barrier and $\beta = \pi/15$, $W = 40$, $w_0 = 10$. Figure 6b shows the packet evolution pattern, manifesting the characteristic trembling motion of ZB with $T_{ZB} = 15$, $A_{ZB} = 2.38$. In the presence of moving barrier with $\upsilon = 1$, $(\Delta\varphi, \Delta A) = (-\pi/2, -\pi/2)$ (Fig. 6d), the packet exhibits ZB outside the barrier and turns into directional transport inside it, restoring to ZB after crossing it, giving rise to $d = -W/(1+\upsilon) = -20$. For $k = -\pi/4$ away from ZB point at $k = 0$ (Fig. 6c), the packet tunnels directly through the barrier, showing a transparent potential rather than the occurrence of ZB. The transparency condition is momentum-independent and fulfilled with broadband feature for any $k \neq 0$, in contrast to above momentum-dependent features in non-relativistic cases. This feature is also rooted in broad linear band nature of Dirac equation. While for $\upsilon = 1/3$ (Fig. 6e), the packet excited at $k = 0$ experiences nearly total reflection at first boundary (Fig. 6f), making ZB and refraction break down. This is because for very small $\beta$, the full reflective case is reached more easily (Fig. S3), leading to the nearly total beam reflection and breakdown of refraction.

## Discussion

In conclusion, we suggested and experimentally demonstrated discrete temporal refractions by a moving gauge-potential barrier on a lattice with broken Galilean invariance. To achieve beam-splitting-free refraction, a quantization condition of potential moving speed $\upsilon$ is revealed, namely $\upsilon$ can only take integer $\upsilon = 1$ or fractional values $\upsilon = 1/q$ (odd $q$). Zero temporal delay is observed for each specific input Bloch momentum, corresponding to directional transparent moving potentials with simultaneous refractionless and reflectionless features. We also demonstrate the refraction effect of *Zitterbewegung* motion by moving potential. Remarkably, in this regime we observe potential-difference-independent temporal delay and momentum-independent transparency by harnessing the linear band nature of Dirac equation. Our work establishes and experimentally demonstrates fundamental laws governing discrete light refraction by moving potentials. This paradigm may find applications in versatile temporal beam steering (see SM. Sec. 5 for typical examples), precise time-delay control and measurement for optical communications and signal processing.

Apart from controlling discrete refraction, moving potentials on a lattice demonstrated in our work can be harnessed to realize many other exotic scattering phenomena rooted in the violation of Galilean invariance. For example, by drifting a disordered potential on a lattice, it is possible to modify the Anderson localization and even wash it out completely with appropriate choice of moving speed[33]. Moreover, by using specially-tailored potentials, such as parity-time-symmetric potentials[51], Kramers-Kronig potentials[39–41], and many others with engineered spatial spectra[61,62] beyond the simplest squared potential barrier, our experimental platform may enable the observation of more exciting discrete-wave mechanics, such as the mass renormalization effects[32], otherwise inaccessible by using stationary potentials in continuous-wave systems.

## Methods

### Experimental setup and measurement techniques

The main experimental setup has been discussed in the main text, let's summarize other experimental details and key measurement techniques. The pulse propagation losses are compensated by EDFAs. To suppress the transient process of EDFA, the signal pulse is mixed with a high-power 1530 nm pilot light before entering EDFA. After EDFA, the pilot light and associated spontaneous emission noise are removed by BPF. PBS and PCs are used to control light polarization in two loops since both MZM and PMs are polarization sensitive devices. ISOs are used to ensure unidirectional circulation in both loops. Key techniques

in our experiments include preparation of a Bloch-wave packet with a required Bloch momentum $k$ and the generation of a sliding gate voltage with a required moving speed $\upsilon$.

### Preparation of a Bloch-wave packet with a required Bloch momentum

A Bloch-wave packet in the temporal lattice corresponds to a pulse train, with amplitude described by $(U, V)^T \exp[-(n - n_0)^2/(\Delta n)^2] \exp(ikn)$, where $\Delta n$ is the Gaussian envelop width and $k$ is the initial Bloch momentum. This pulse train is generated from a preliminary evolution of a single pulse injected from the long loop to guarantee the coherence for interference purpose at the coupler. During the preliminary evolution, the pulses circulating in the short loop are attenuated every other round trip by MZM while those in the long loop are kept constant in each round trip[54]. With this approach, we can obtain a pulse train with $\Delta n \approx 10$ after $-m = 100$ circulation times. The Bloch momentum of packet is $k = (\pi-\phi)/2$, where $\phi$ is the short-loop phase modulation in each round trip. Then, by applying appropriate phase and intensity modulation in the 101st step, the required eigenstate $(U, V)^T$ can be imparted to the packet.

### Generation of a sliding gate voltage with a required moving speed $\upsilon$

The moving gauge-potential barrier is created by controlling the modulation waveform generated from AWGs. Let's take $\upsilon = 1$ and $\upsilon = 1/3$ as examples. For $\upsilon = 1$, the sliding gate voltage (with a real width of $W\Delta t$) needs to delayed exactly by one lattice site during one evolution step, corresponding to a time delay of $\Delta t$ in each modulation period $T$. While for $\upsilon = 1/q$, $(q = 3, 5,...)$, this speed is an average speed, characterizing the averaging site number the gate voltage moves in one period. For example, $\upsilon = 1/3$ can be realized by introducing a time delay $\Delta t$ in every three periods $3T$. Since $T/\Delta t = 25\mu s/75$ ns ~ 333, the fiber loop can accommodate 333 evolution steps, which are enough to observe the refraction effect.

### Reporting summary

Further information on research design is available in the Nature Portfolio Reporting Summary linked to this article.

## Data availability

The data supporting the findings of this study are available within the paper and its Supplementary Information files. The data generated in this study are provided in Supplementary Information/Source Data file. Source data are provided with this paper.

## Code availability

The codes used to obtain the main Figures in the main text and Supplementary Materials are provided in Supplementary Information/Source data file, together with the source data.

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

## Acknowledgements

The work was supported by fundings from the National Natural Science Foundation of China (12374305 (B.W.), No. 12204185 (C.Q.), 11974124 (B.W.), 62305122 (S.W.), and 12021004 (P.L.)), and Natural Science Foundation of Hubei Province (No. 2022CFB036 (C.Q.)).

## Author contributions

C.Q. and B.W. conceived the idea. C.Q., B.W., H.Y., and S.W. designed the experiment. C.Q., H.Y., L.Z., and S.W. performed the experiment. M.L., Y.L., X.H., and C.L. assisted the experiment. C.Q. and H.Y. analyzed the data. C.Q. and S.L. provided the theoretical support. B.W. and P.L. supervised the project. All authors contributed to the discussion of the results and writing of the manuscript.

## Competing interests

The authors declare no competing interests.
