## [Peer Review File · Nature Communications]

REVIEWER COMMENTS

Reviewer #1 (Remarks to the Author):

The manuscript entitled "Observation of discrete-light temporal refraction by moving potentials with broken Galilean invariance" describes theoretically and experimentally a 2d space time lattice with temporal modulation obtained from a system of coupled fibre loops with a varying phase delay. The main result is quantisation of velocities that allow transmission, reflection-free for specific modulations, dynamic localisation and Zitterbewegung refraction with weak coupling.

The manuscript is detailed, and address the field of light scattering by moving potential, and time-varying metamaterials, which is of great importance for the community of optics and wave science. This work is solid with very interesting results, but the presentation is difficult to follow and sometimes obscure; figures are not properly introduced, and it is not clear what is the storyline, and the text is full of details which don't link with each other, more akin to a lab book than a paper. After reading the paper a few times, I am still not able to follow the main claims.

I believe this is an important contribution, therefore I would like to recommend it for publication in Nature Communications, but in the present state I don't think it is clear enough for the broad readership of the journal.

. The authors need to explain better in the introduction why Galilean invariance is more relevant than Lorentzian despite relativistic speeds involved.

. Can the authors describe better the physical origin of the lattice? presenting the setup first would simplify the understanding, I don't think it is useful as figure 6

. Using $v=1$ in fig 1 might cause misunderstanding, use $1/3$ from SI perhaps?

. Should the Zitterbewegung methods be in the main text? I find it very technical, maybe it can be moved to the SI.

. Typo: line 291 "simply"

Reviewer #2 (Remarks to the Author):

The authors present an extensive analysis of a fibre loop system, that can be employed to realise discrete temporal refraction. In their work, using the fact that, unlikely their continuous counterparts, discrete waves are not covariant under Galilean boosts, the authors show temporal refraction by a moving gauge potential barrier.

The work is of high quality, well-written, and well-communicated. I personally think that this work is of high-enough impact to deserve publication in Nature Communications, as it has the potential for new and interesting research, based on Galilean invariance symmetry breaking in discrete systems. Fibre

loops, in fact, have been proven to be a rich and interesting platform, where to study, amongst other things, scattering from time-dependent potential and other time-dependent related phenomena. Having at disposal a complete framework, like the one presented by the authors, dealing with discrete temporal refraction in such structures, will surely allow us to achieve a better understanding of photon dynamics in such systems, and open the way for novel applications, like those mentioned by the authors in the Discussion Section, for example.

For these reasons, I would suggest to accept this manuscript for publication in Nature Communications as it is. Its clarity of exposition and extensive Methods section make it very appealing to the expert, as well as to the non expert reader.

Response Letter of manuscript

Reviewer #1 (Remarks to the Author):

The manuscript entitled "Observation of discrete-light temporal refraction by moving potentials with broken Galilean invariance" describes theoretically and experimentally a 2d space time lattice with temporal modulation obtained from a system of coupled fiber loops with a varying phase delay. The main result is quantization of velocities that allow transmission, reflection-free for specific modulations, dynamic localization and Zitterbewegung refraction with weak coupling.

The manuscript is detailed, and address the field of light scattering by moving potential, and time-varying metamaterials, which is of great importance for the community of optics and wave science. This work is solid with very interesting results, but the presentation is difficult to follow and sometimes obscure; figures are not properly introduced, and it is not clear what is the storyline, and the text is full of details which don't link with each other, more akin to a lab book than a paper. After reading the paper a few times, I am still not able to follow the main claims.

I believe this is an important contribution, therefore I would like to recommend it for publication in Nature Communications, but in the present state I don't think it is clear enough for the broad readership of the journal.

Reply: We thank the reviewer for her/his positive assessment of our work and for the valuable comments aimed at improving the quality of presentation. The main claim of this paper is to establish and demonstrate new laws for governing discrete time refractions by a moving potential barrier with broken Galilean invariance. The storyline (logic) of the paper has been now clarified in the revised manuscript. We firstly introduce the synthetic temporal lattice system and show that the discrete refraction by a moving potential barrier on the lattice depends on the drift velocity owing to Galilean invariance breakdown (Section 1). Then, we identify the selection rules for potential moving speed v to ensure a well-defined refraction (Section 2). This leads to a quantization condition for the moving speed $v = 1/q$ (odd q) to ensure a beam-splitting-free refraction. In what follows, we choose two representative permitted integer and fractional moving speeds to experimentally demonstrate discrete

refraction by a moving potential barrier (Section 3). This includes measurements of beam delays and demonstration of the transparency condition. In Section 4, we generalize the static moving potential to a time-modulated moving one, with modified refraction properties, beam delay and transparency condition. Finally, we consider the relativistic limit of the discrete dynamical equations and reveal unique refraction effects in this regime (Section 5).

To make the storyline clearer and the presentation of the paper more fluent and less obscure, we revised the titles for all sections of Results part and rewrote the first paragraph of each section to clarify their logic and connections among all sections. All revisions are highlighted in red color in the amended manuscript version. Finally, in order to make the paper more compact and the main message more striking and clearer, we condensed all the technical details throughout the paper, moving some details into the Supplementary Materials and Methods part.

(1) The authors need to explain better in the introduction why Galilean invariance is more relevant than Lorentzian despite relativistic speeds involved.

Reply: We thank the Referee for raising such an interesting question. The reason why Galilean invariance, rather than Lorentz invariance, is more relevant in our system is that, even though light pulses travel in the two fiber loops at the speed of light $c/n_g \sim$ at c scale, the effective light dynamics and refraction in the lattice involves non-relativistic speeds and can be described, in the continuous space-translational invariance limit of the lattice, by a Schrodinger equation displaying Galilean (rather than Lorentz) invariance. The spreading speed of the optical pulses in the synthetic lattice is determined by the lattice's group velocity v_g . Unlike the physical phase or group velocity of the pulse, which is of the order of magnitude of light speed in vacuum, the lattice's group velocity, $v_g = \partial \varepsilon(k)/\partial k$, which denotes how many lattice sites are travelled by the wave packet per unit time, is much smaller than c and is of the order of $v_g \sim c(\Delta t/T) \ll c$, where $\Delta t/T = \Delta L/L$ is related to the fiber length unbalance of the two loops (in our experimental condition $\Delta t/T = 75\text{ns}/25\mu\text{s} = 1/333$). Likewise, the potential barrier moving speed v , describing how many sites the potential moves per unit time step, is of the order of v_g and it is thus a non-relativistic speed. It is the non-parabolic nature of dispersion relations in lattice systems (rather than relativistic speeds involved in the dynamics) the reason why Galilean invariance breaks down. To better clarify this point and the physical relevance of Galilean invariance, we revised paragraph 2 of introductory part, relevant explanations of Eq. (3) and paragraph 1, Section 5 of Results

part, see the revised text highlighted in blue color.

(2) Can the authors describe better the physical origin of the lattice? presenting the setup first would simplify the understanding, I don't think it is useful as figure 6.

Reply: We thank the reviewer for this comment and suggestion. In the revised version, we expanded paragraph 1, Section 1 of Results part, where the physical origin of the synthetic temporal lattice using two fiber loops has been described. Basically, the pulse physical time is decomposed as $t_n^m = mT + n\Delta t$, where $T = L/c_g$ is mean travel time and $\Delta t = \Delta L/c_g \ll T$ is travel-time difference in two fiber loops, $c_g = c/n_g$ is pulse's group velocity, $n_g = 1.5$ is the group index in fiber and c is vacuum light speed. The pulse dynamics can thus be mapped into a "link-node" lattice model (n, m) [see Fig. 1c in the revised manuscript], where n, m denote the transverse lattice site and longitudinal evolution step. The leftward/rightward links towards the node correspond to pulse circulations in short/long loops and scattering at each node corresponds to pulse interference at the coupler.

To introduce the schematic experiment setup firstly, Figs. 1a and 1b have been exchanged with Figs. 1c and 1d. Also, the original Fig. 6 of explaining experimental setup and measurement techniques have been moved to the new Fig. 2. We also added a paragraph in Section 3 after Eq. (9) of Results part to introduce experimental setup and key measurement techniques. More experimental technical details are discussed in the Methods part.

(3) Using $v=1$ in fig 1 might cause misunderstanding, use $1/3$ from SI perhaps?

Reply: We thank the Referee for this pertinent suggestion, with which we fully agree. The potential moving speed $v = 1/3$ is indeed more representative than $v = 1$ since v can only take the specific integer number $v = 1$ but a series of fractional numbers $v = 1/q$ (odd q). According to the Referee's suggestion, in the revised version, we exchanged $v = 1/3$ case from SI with $v = 1$ case in Fig. 1 of the main text.

(4) Should the Zitterbewegung methods be in the main text? I find it very technical, maybe it can be moved to the SI.

Reply: Thanks for this pertinent suggestion. According to the Referee's recommendation, we have moved the derivation of Dirac equation and the whole Zitterbewegung methods part to Section 4 of SI.

(5) Typo: line 291 “simply”

Reply: we have revised “simply” to “simplify”.

Reviewer #2 (Remarks to the Author):

The authors present an extensive analysis of a fibre loop system, that can be employed to realize discrete temporal refraction. In their work, using the fact that, unlikely their continuous counterparts, discrete waves are not covariant under Galilean boosts, the authors show temporal refraction by a moving gauge potential barrier.

The work is of high quality, well-written, and well-communicated. I personally think that this work is of high-enough impact to deserve publication in Nature Communications, as it has the potential for new and interesting research, based on Galilean invariance symmetry breaking in discrete systems. Fibre loops, in fact, have been proven to be a rich and interesting platform, where to study, amongst other things, scattering from time-dependent potential and other time-dependent related phenomena. Having at disposal a complete framework, like the one presented by the authors, dealing with discrete temporal refraction in such structures, will surely allow us to achieve a better understanding of photon dynamics in such systems, and open the way for novel applications, like those mentioned by the authors in the Discussion Section, for example.

For these reasons, I would suggest to accept this manuscript for publication in Nature Communications as it is. Its clarity of exposition and extensive Methods section make it very appealing to the expert, as well as to the non-expert reader.

Reply: We thank the Referee for her/his very positive evaluation of our work, recommending its publication in Nature Communications as it is.

REVIEWERS' COMMENTS

Reviewer #1 (Remarks to the Author):

The authors have done a good job at clarifying the manuscript. I still find it quite dense, but I think it is clear enough for a technical readership.

Also, all specific queries have been answered satisfactorily.

I therefore recommend it for publication with Nature Communications.

Response Letter of manuscript

Reviewer #1 (Remarks to the Author):

The authors have done a good job at clarifying the manuscript. I still find it quite dense, but I think it is clear enough for a technical readership.

Also, all specific queries have been answered satisfactorily.

I therefore recommend it for publication with Nature Communications.

Reply: We thank the reviewer for her/his positive assessment on our work and for recommending it for publication on Nature Communications.